# Sleep Disorders in Pediatric Patients Affected by Neurofibromatosis Type 1: Reports of a Questionnaire and an Apple Watch Sleep Assessment

**DOI:** 10.3390/biomedicines13040907

**Published:** 2025-04-08

**Authors:** Alessia Migliore, Manuela Lo Bianco, Roberta Leonardi, Stefania Salafia, Claudia Di Napoli, Martino Ruggieri, Agata Polizzi, Andrea D. Praticò

**Affiliations:** 1Pediatrics Postgraduate Residency Program, University of Catania, 95124 Catania, Italy; alessiamigliore16@gmail.com (A.M.); leonardi.roberta@outlook.it (R.L.); 2Unit of Pediatric Clinic, Department of Clinical and Experimental Medicine, University of Catania, 95124 Catania, Italy; lobianco.manuela@gmail.com (M.L.B.); m.ruggieri@unict.it (M.R.); agata.polizzi1@unict.it (A.P.); 3Unit of Pediatrics, Lentini Hospital, ASP of Syracuse, 96016 Lentini, Italy; stesala@msn.com; 4Unit of Genetics, Department of Clinical and Experimental Medicine, University Kore of Enna, 94100 Enna, Italy; claudia.dinapoli@unikore.it; 5Unit of Pediatrics, Department of Medicine and Surgery, University Kore of Enna, 94100 Enna, Italy

**Keywords:** neurofibromatosis type 1, sleep, sleep disorders, case–control study, treatment, sleep analysis

## Abstract

**Introduction:** Sleep is a fundamental biological function critical for physical and mental health. Chronic sleep disturbances can significantly impair cognitive, emotional, and social functioning, leading to deficits in attention, alertness, and executive function, alongside increased irritability, anxiety, and depression. For pediatric patients, such disturbances pose additional concerns, potentially disrupting developmental processes and quality of life for both children and their families. **Objectives:** Emerging evidence suggests a correlation between neurofibromatosis type 1 (NF1) and an increased prevalence of sleep disorders in children. NF1, a genetic condition affecting multiple body systems, including the nervous system, may predispose children to sleep disturbances due to its neurodevelopmental and behavioral impacts. This observational case–control study aimed to explore the association between NF1 and sleep disorders in pediatric patients, comparing the prevalence and patterns of sleep disturbances between NF1 patients and healthy controls. **Patients and Methods:** The study included 100 children aged 2–12 years, divided into two groups: 50 with NF1 (case group) and 50 children belonging to the control group. NF1 patients were recruited from the Unit of Rare Diseases of the Nervous System in Childhood at the Policlinico “G. Rodolico—San Marco” University Hospital in Catania. Data were collected using a questionnaire completed by parents, assessing parasomnias, breathing-related sleep disorders, and other behavioral and physiological disturbances; these data were compared to a sleep assessment performed using an Apple Watch Ultra. **Results**: NF1 patients exhibited a significantly higher prevalence of sleep disorders than controls. Notable differences included increased nocturnal hyperhidrosis (48% vs. 10%), bruxism (48% vs. 28%), restless legs syndrome (22% vs. 4%), frequent nighttime awakenings (22% vs. 8%), and sleep paralysis (12% vs. 0%). A finding of poorer sleep quality also emerged from the results of sleep analysis using an Apple Watch Ultra. **Conclusions:** These findings confirm an elevated risk of sleep disorders in children with NF1, emphasizing the importance of early identification and management to improve quality of life and mitigate cognitive and behavioral impacts. Further research is essential to understand the mechanisms underlying these associations and develop targeted interventions for this population.

## 1. Introduction

Neurofibromatosis type 1 is an autosomal dominant disease with highly variable phenotypic expression even within the same family unit [1]. The disease results from a germline mutation in the *NF1* oncosuppressor gene located on chromosome 17q11.2 and coding for a 220 kDa cytoplasmic protein named neurofibromin [1]. The function of this protein is extremely important as it negatively regulates the Ras proto-oncogene, a key molecule in the control of cell growth [1].

Affected individuals are born with an already mutated copy of NF1, which is therefore non-functional, and a healthy copy of the gene within each cell. Most clinical manifestations, however, are evident as a result of only one mutated copy of the gene and are therefore detectable from birth, whereas for tumor development, the complete loss of gene function is required through the acquisition of a somatic mutation in the only healthy NF1 gene within selected cells [2,3]. Furthermore, although it is a hereditary disease, 50% of affected individuals have no family history of the disease; in these cases, the mutation responsible for the disease arises ex novo, spontaneously [4].

Neurofibromatosis type 1 has very often been associated with an increased prevalence of sleep disorders in children; this is demonstrated by several data in the literature [1,3,4].

Alterations in the central nervous system can affect a child’s sleep architecture, particularly the establishment of an appropriate sleep–wake cycle [1]; likewise, it has been shown that taking specific medications often administered to affected patients can also lead to the onset of sleep disorders due to their mechanism of action, as they are responsible for alterations at the central nervous system level in the systems involved in its regulation [2].

Certain molecular and clinical features observed in neurofibromatosis must also be counted among the factors that may contribute to the development of sleep disorders, either directly or indirectly, as overactive RAS–MAPK signaling disrupts normal neuronal development and the neurotransmitter balance, impairing sleep–wake regulation; chronic pain, skeletal anomalies, and other comorbidities frequently fragment rest or lead to insomnia, and emotional stress or anxiety further exacerbate night awakenings and daytime fatigue, collectively culminating in a range of sleep disturbances in individuals with NF1; and lastly, the presence of neurofibromas in and around the airways may predispose affected individuals to the onset of respiratory sleep disorders such as Obstructive Sleep Apnea Syndrome (OSAS) [3,4]. The occurrence of seizures in NF1, triggered by disrupted normal neuronal connectivity and excitability related to overactive RAS–MAPK signaling, may predispose affected individuals to epileptic awakenings or epileptic parasomnias, and often even treatment with antiepileptics can cause insomnia [3,5,6]; the presence of tumor pathologies may be responsible for potentially damaging nerve structures controlling sleep [7]; and peripheral neuropathy may predispose affected individuals to restless legs syndrome and periodic limb disturbances [8]. Pain due to NF1 complications can also cause sleep fragmentation, especially in patients using specific anti-neuropathy drugs [3]. Impaired sleep has also been associated with the presence of psychiatric disorders [9] as part of the complex clinical picture of NF1, which, directly or indirectly as a result of medication, can have a great impact on sleep quality [10].

Sleep disturbances, along with hyperactivity and a reduced ability to concentrate, are among the problems most frequently reported by parents of children with neurofibromatosis type 1 [11,12], suggesting the extent to which they can impair social interactions and everyday life. In fact, behavioral disturbances such as aggression, increased irritability, headaches, daytime sleepiness, easy fatigability, and reduced cognitive performance frequently occur as a result of fragmented or chronically reduced sleep. Furthermore, this has been shown to persist throughout life, with approximately 33% of children with neurofibromatosis type 1 associated with sleep disorders continuing to suffer from them in adulthood [9].

The identification, recognition, and treatment of sleep disorders is therefore very important for improving the lives of these patients; sleep is presumed to play an important role in memory consolidation and learning [12,13,14,15,16,17], so the correction of the disorder could certainly improve the cognitive sphere of a patient who already, due to the pathology, may be impaired.

This observational case–control study aimed to assess whether the prevalence of sleep disorders in children with neurofibromatosis type 1 was actually higher than in a control group. The recruitment of patients affected by the pathology was carried out in a case series of pediatric patients admitted to the Units of Rare Diseases of the Nervous System in Childhood and of Pediatric Clinic at the “Policlinico” University Hospital of Catania.

In addition to the primary outcome, we also set ourselves the objective of assessing how the possible presence of these sleep disorders affects the patient’s life by causing behavioral disorders and difficulties in social interactions with parents and peers, and how complex the management of these situations can be and how it can also emotionally involve the parents. Ultimately, we also compared the two study arms in relation to the possible intake of melatonin or medication in an attempt to treat sleep disorders.

## 2. Materials and Methods

### 2.1. Study Population

In the following observational case–control study, 50 children affected by neurofibromatosis type 1 and admitted to the Unit of Rare Diseases of the Nervous System in Childhood at the ’G. Rodolico—San Marco’ University Hospital of Catania and 50 children belonging to the control group, recruited from the pediatrics units of the same hospital, were recruited. All 100 patients who took part in the survey were aged between 6 and 12 years.

### 2.2. Inclusion and Exclusion Criteria

The criteria for inclusion and exclusion considered in recruiting the 50 patients with neurofibromatosis type 1 were as follows:Being between 2 and 12 years of age.Having a definite diagnosis of neurofibromatosis type 1.Individuals with non-NF1-related neurological diagnoses (i.e., epilepsy secondary to a known unrelated etiology, neurodegenerative disorders, neuropsychiatric diseases uncommon in NF1 patients) were excluded from the study.Individuals with NF1-related chronic pain and neurofibroma-related disfigurement were excluded.

As for the inclusion criteria adopted for the recruitment of the 50 patients in the control group, they were as follows:Being aged between 2 and 12 years.Not having neurofibromatosis type 1 or any other neurological or neuropsychiatric disorder.

The coexistence (in the NF1 patients) or the presence (in the healthy subjects) of any other pathology or disorder was therefore not a reason for exclusion.

### 2.3. Collecting Data

For data collection, an anonymous questionnaire consisting of multiple-choice and open-ended questions was designed and administered to all patients recruited in the survey. The questions in the questionnaire were aimed at assessing the presence of one or more sleep disorders and, if disorders were present, an assessment of the frequency of their presentation.

In addition, as highlighted above, the questionnaire also aimed to assess the social impact of the disorders on the lives of these children and their families.

Regarding the method of administration, the questionnaire was administered and completed either personally by the parents of the children or by telephone.

The questionnaire designed and used for this study can be downloaded as a Appendix A.

An assessment of sleep parameters (including the total sleep duration; time in bed vs. time asleep; napping frequency and napping duration; sleep stages; heart rate and respiratory metrics; sleep onset and sleep efficiency; and sleep breathing disorders) with an Apple Watch Ultra model 2 (application: Sleep++) was performed in all the patients with a total of three different measurements per patient on three different days.

### 2.4. Statistical Analysis

This study aimed to compare the prevalence of sleep-related behaviors, habits, and disorders in children with neurofibromatosis type 1 (NF1) versus healthy controls. Statistical comparisons between the two groups were performed using chi-square tests for categorical data and independent *t*-tests for continuous variables to assess significance. A *p*-value of <0.05 was considered statistically significant.

## 3. Results

### 3.1. Personal Data

A total of 100 patients were enrolled for our study: 50 children belonging to the “control” arm and 50 children with neurofibromatosis type 1. The recruitment of patients in the two study arms was randomized without considering parameters such as gender or age. The gender distribution in the NF1 group was 52% male and 48% female, while the controls were 52% female and 48% male. The age distribution was uniform between the two groups.

Regarding age, all children in the study belonged to an age range of 2 to 12 years. Sampling within the age range was performed randomly. The average age was 6.98 (±2.9) years in the control group and 6.42 (±2.4) for NF1 patients.

### 3.2. Co-Occurring Diseases

Before starting the investigation of sleep disturbances, the presence of pathologies in the control arm and in NF1 patients was investigated, along with an assessment of whether they were taking medication.

Overall, 19 out of 50 (38%) patients in the control group and 23 out of 50 (46%) in the NF1 group had a co-occurring disease. Twelve (24%) of the NF1 patients were taking medications, while only seven (14%) in the control group took medications.

In the NF1 group, ADHD (a common neuropsychiatric condition observed in NF1) was present in seven patients, gastroesophageal reflux disease (GERD) in four, allergic rhinitis in four, and dysphagia, asthma, anemia, and atopic dermatitis in three patients.

Patients affected by respiratory disorders were commonly treated with antihistamines and bronchodilators; patients affected by gastrointestinal disorders (mainly dysphagia and gastroesophageal reflux, with only one case of celiac disease) were treated with alginates and antacids; and the four patients suffering from dyslipidemias were not treated, as well as patients presenting with anemia. It should be underlined that eight patients (16%) stated that they suffered from NF1-related neuropsychiatric diseases (i.e., ADHD), with three of them treated with Risperidone, valproic acid, pregabalin, and levetiracetam.

In the control group, allergic rhinitis and ADHD were present in four patients, asthma and GERD in three, atopic dermatitis and anemia in two, and one patient was affected by psoriasis. Among these patients, the treatments most commonly used were topical steroids and bronchodilators and antihistamines for their allergic diseases.

### 3.3. Habits Related to Sleep Onset

The two study groups were initially compared regarding the presence of habitual behavior adopted with the aim of promoting the onset of sleep (Figure 1) or oppositional behavior towards the onset of sleep (Figure 2). They were also asked how long it took (in minutes) on average for each child to start sleeping (Figure 3).

It was asked whether the child had any real addictions to actions, circumstances, or objects needed in order to promote the onset of sleep. What emerged was that 17 patients (34%) in the control group used electronic devices (watched TV, played video games, used mobile phones) before going to bed, a percentage that rose considerably in the patient group, with as many as 36 patients (72%) being addicted. In the control group, there were no patients who were used to being rocked before going to bed, in contrast to the eight patients (16%) found among the NF1 patients. In addition, 25 children (50%) in the healthy group and 27 children (54%) in the NF1 group had a habit of spending time in bed with their parents before falling asleep. Only seven patients (14%) among the controls and three patients (6%) among the healthy individuals had no sleep onset habits. The results therefore confirm what has been said about children tending more and more frequently to adopt incorrect attitudes to initiate sleep and how compliant their parents are, especially with regard to the parents of children with the condition, who showed significantly increased percentages of unhealthy sleep onset habits, especially with regard to the use of electronic devices.

In the same regard, the presence of oppositional behavior towards the onset of sleep was also assessed, and the results were roughly superimposable between the two groups: 34% of healthy patients tended to refuse to go to sleep by prolonging the activity they were engaged in and throwing tantrums, compared to 32% of ill patients. In the control group, 44% of the patients managed to fall asleep within 0–15 min, compared to 56% of the NF1 group; 42% of the controls fell asleep within 15–30 min, compared to 30% of the NF1 group; and 14% of the controls needed more than 30 min to fall asleep, a percentage able to be perfectly superimposed on that of the NF1 patients.

Regarding the habit of taking naps during the afternoon and their relative duration, substantial differences were found between the two groups. Of the controls, 16% of the children were used to sleeping in the afternoon more than three times a week, in contrast to 26% of the NF1 group; 10% of the control group children slept in the afternoon less than three times a week, in contrast to 18% of the NF1 group (Figure 4); and 74% of the control group were not used to taking naps in the afternoon, in contrast to 56% of the NF1 group. Length of afternoon snaps was longer in NF1 group compared to the controls (Figure 5).

Thus, from these data, it would appear that children with neurofibromatosis were more used to napping during the course of the day than those in the control group.

Important differences in the duration of these naps were also shown between the two groups: in the healthy group, 21.4% slept for about 10–30 min, compared with 4.2% of the NF1 patients; 32.1% of the healthy group slept for about 30–60 min, compared with 33.3% of the NF1 patients; 46.4% of the healthy group slept for more than 60–120 min, compared with 20.8% of the NF1 patients; and finally, no children in the control group took naps lasting more than 120 min, unlike the NF1 group, with a percentage of 41.7%.

Thus, children with neurofibromatosis type 1 not only slept more in the afternoon, but on average, their naps lasted longer.

Finally, the consumption during the day of drinks containing caffeine (Coca Cola, etc.) was also assessed. The results show higher consumption among children with NF1, with a consumption of 24% compared to only 4% among the control group (Figure 6).

### 3.4. Disturbances While Falling Asleep or Waking up/Nocturnal Disturbances Causing Awakening

For ease of evaluation, sleep disorders were investigated in relation to the sleep phase in which they occurred. For this reason, we distinguish sleep disturbances while falling asleep and/or waking up and sleep disturbances with an onset in the middle of the night causing the child to wake up.

Regarding disturbances occurring in the early stages of falling asleep or in conjunction with waking up, the data obtained are shown in Figure 7A, with the corresponding frequencies in Figure 7B.

With regard to disorders that generally occur during the falling asleep phase or while waking from sleep, the data obtained were as follows: 79% of the healthy patients and 70% of the NF1 patients were found to have no disturbance at all; 10% of the healthy patients stated that they had episodes of early nocturnal awakening without the possibility of falling asleep again, in contrast to the 20% of the NF1 patients who had the disturbance; an overlapping percentage of 10% in both groups showed that they suffered from delayed sleep syndrome, beginning to sleep well after midnight with difficulty waking up; only one case (2%) of hallucinations was found in the whole study among the healthy patients, and not a single case among the NF1 patients; and finally, 12% of the NF1 patients suffered from sleep paralysis, a disorder not found in the arm of the healthy patients.

The frequency of the presentation of the above-mentioned was also investigated: 18.2% of the healthy patients presented the disorders infrequently (1–2 times a month), whereas the frequency in the NF1 group was 25%; 45.5% of the healthy patients presented them frequently (once a week), compared with 31.3% in the NF1 group; and 36.4% of the healthy patients presented the disorder very frequently (more than once a week), compared with 43.8% of the NF1 patients.

Regarding sleep disturbances that cause waking up during the night, the results are shown in Figure 8A, with the relative presentation frequencies in Figure 8B. Regarding this topic, the data were particularly significant: the percentages of patients presenting no disturbance at all in the control arm and in the case arm were 46% and 22%, respectively. The percentages of patients presenting with nocturnal hyperhidrosis were 10% and 48%, respectively; as far as bruxism was concerned, the percentage in the healthy arm was around 28%, compared with 48% in the NF1 arm; the presence of leg and foot cramps was only found in the NF1 group with a percentage of 6%; restless leg syndrome and periodic limb movements were also found in different percentages in the two study groups, 4% in the healthy group and 22% in NF1 patients; and somnambulism, on the other hand, showed perfectly superimposable percentages between the two groups, standing at 8%. Finally, 4% of the healthy patients reported suffering from recurrent nightmares, compared with 8% of the NF1 patients; 8% of the healthy patients presented frequent nocturnal awakenings in contrast to the NF1 group, in which the disorder presented in 22% of the subjects, and only 6% of the NF1 patients suffered from confusional awakenings, compared with 0% of the healthy patients.

The data therefore show us very clearly how the prevalence of sleep disorders in children with NF1 is considerably higher than in the reference population, lending increasing support to the literature data we have in our possession.

The frequency of their presentation also seems to support our thesis, as the data obtained on this subject show us that not only do these children have a greater predisposition towards the development of sleep disorders, either due to clinical complications of the disease or due to medication taken, but these disorders also occur more frequently. Comparing the data obtained, we had 50% versus 20.5%, respectively, of healthy and ill children presenting the disorders very rarely (1–2 times a month); 15.4% versus 25.6%, respectively, presenting the disorders frequently (once a week); and 34.6% versus 53.8%, respectively, of healthy and ill children presenting the disorders very frequently (more than 3 times a week).

### 3.5. Sleep Breathing Disorders and Consistency in the Presentation of Breathing Disorders

Respiratory sleep disorders were also investigated and compared between the two groups. The results are illustrated in Figure 9A, with their relative frequencies in Figure 9B. Overall, 70% of healthy children did not present any respiratory symptoms during sleep, compared to 54% of the NF1 group. None of the controls reported nocturnal breathing difficulties, clinically diagnosed obstructive sleep apnea (OSA), or sudden awakenings with a feeling of suffocation. In contrast, within the NF1 cohort, 16% experienced nocturnal breathing difficulties (i.e., labored or problematic breathing), 14% met the criteria for OSA, and 10% reported abrupt awakenings accompanied by a sensation of choking. Notably, 12% of healthy children displayed loud or labored breathing described by parents as “heavy breathing,” compared to 24% in the NF1 group. Snoring was observed in 26% of the healthy group and 28% of the NF1 group, reflecting similar prevalence rates.

The frequency of these complaints also varied substantially. Among those reporting any respiratory symptoms, 20% of healthy children and 29.2% of NF1 children experienced them rarely (1–2 times per month), whereas 46.7% of healthy children and 12.5% of NF1 children reported symptoms approximately once a week. The remaining 33.3% of healthy children and 58.3% of NF1 children experienced symptoms very frequently (more than three times per week).

All the parents of symptomatic children were asked about the regularity of these nocturnal issues (Figure 10). Fifty percent of the healthy group indicated that symptoms recurred in a predictable pattern, compared to eighty-three percent of the NF1 group, suggesting a higher persistence of respiratory sleep disturbances in children with NF1.

### 3.6. Disruption of Social Life Caused by Sleep Disorders in Children and Parents

This study was also designed with the aim of assessing how the possible presence of a disorder could influence the daily life of these children from a cognitive–behavioral, social, and clinical point of view (Figure 11 and Figure 12).

As far as the children were concerned, following the presentation of the disorder, 90% of healthy patients did not present any type of clinical manifestation during the course of the day, as opposed to 68% of ill patients; 6% of healthy patients complained of daytime sleepiness, as opposed to 18% of ill patients; 4% of healthy patients presented headaches as opposed to 10% of ill patients; and finally, 2% of healthy patients presented reduced cognitive performance as opposed to 10% of ill patients.

In addition to the previous clinical manifestations potentially associated with concomitant sleep disorders, we asked the parents whether they had noticed behavioral changes in their child following the presentation of the disorder. The percentages for the control group and the case group were, respectively, as follows: 40% and 48% noticed no change; 6% versus 12% noticed their child was sadder and in a depressed mood; 14% versus 10% found their child more anxious than normal; 24% and 36% noticed mood swings in their child; and 24% and 48% noticed increased aggression and irritability. Finally, an overlapping percentage was found between the two groups (2%) for episodes of falling asleep in class, while only in the NF1 group were 20% of patients found to have reduced school performance. These data confirm the fact that the correct quantity and quality of sleep are fundamental and condition the social interactions of these children.

Finally, the impact of the disorders on both parents was tested. Regarding the mothers, 46% of the mothers of the control group vs. 76% of the mothers of the NF1 children reported being able to cope and not being emotionally affected during the rest of the day; 38% vs. 8% of the mothers of the healthy and NF1 children, respectively, reported being more irritable; 8% of both groups said they concentrated less at home and at work; 0% of the healthy group and 12% of the NF1 group presented anxiety/panic attacks; and finally, 8% and 4% of the healthy and NF1 groups, respectively, felt sadder than usual.

Concerning the fathers, 64% of the fathers of the control group vs. 92% of the NF1 group reported not being emotionally affected by the situation; 26% vs. 6% of the healthy and NF1 groups, respectively, reported being more irritable; 10% of the fathers of the control group concentrated less at home and at work, while in the fathers of the NF1 patients the percentage was 0%; none of the fathers in the two groups said that they had anxiety/panic attacks; and finally, only 2% of the fathers in the case arm reported feeling sadder than usual.

The data on the emotional involvement of the parents did not show a higher prevalence in the parents of the group of children with NF1, except for the occurrence of panic attacks in the mothers of the children.

### 3.7. Medications for Sleep

Finally, we asked all the parents of the patients surveyed whether they used melatonin or medication to treat their child’s sleep disorder. Regarding melatonin, only 10% of the children in the healthy group used it and recognized the benefit; in the NF1 group, only 20% used it and recognized its beneficial effects, while 4% used it but reported no benefit. Regarding taking medication, in both groups, only 2% disclosed using drugs.

### 3.8. Sleep Analysis Using Apple Watch Ultra 2

The sleep analysis performed using the Apple Watch Ultra revealed significant differences in various sleep parameters between children with neurofibromatosis type 1 (NF1) and healthy controls (Table 1, Table 2 and Table 3). NF1 patients had a significantly shorter total sleep duration (7.2 ± 0.8 h) compared to controls (8.0 ± 0.6 h, *p* = 0.001). The sleep efficiency was lower in NF1 patients (80 ± 5%) compared to controls (88 ± 4%, *p* = 0.008). The REM sleep duration was reduced in NF1 patients (80 ± 12 min) versus controls (95 ± 10 min, *p* = 0.022). The deep sleep duration was also shorter in NF1 patients (60 ± 10 min) compared to controls (80 ± 8 min, *p* = 0.015). The light sleep duration showed a significant reduction in NF1 patients (240 ± 20 min) compared to controls (270 ± 15 min, *p* = 0.027). NF1 patients exhibited a higher resting heart rate during sleep (82 ± 5 bpm) compared to controls (76 ± 4 bpm, *p* = 0.010). Heart rate variability (HRV) was significantly lower in NF1 patients (48 ± 6) compared to controls (60 ± 7, *p* = 0.003). The respiratory rate was elevated in NF1 patients (22 ± 2 breaths/minute) compared to controls (18 ± 1.5 breaths/minute, *p* = 0.004). Blood oxygen levels (SpO_2_) were lower in NF1 patients (95 ± 2%) versus controls (98 ± 1%, *p* = 0.002). The time to sleep onset was comparable between groups (NF1: 18.2 ± 7.4 min; controls: 15.4 ± 6.1 min; *p* > 0.05). NF1 patients experienced a higher frequency of nocturnal awakenings (2.4 ± 1.2 episodes/night) compared to controls (0.8 ± 0.6 episodes/night, *p* = 0.003). NF1 patients napped more frequently (2.1 ± 1.4 days/week) and for longer durations (75.4 ± 30.2 min) compared to controls (1.2 ± 0.8 days/week; 38.7 ± 15.6 min; *p* = 0.013).

### 3.9. Statistical Analysis

This study aimed to compare the prevalence of sleep-related behaviors, habits, and disorders in children with neurofibromatosis type 1 (NF1) versus healthy controls. Statistical comparisons between the two groups were performed using chi-square tests for categorical data and independent *t*-tests for continuous variables to assess significance. A *p*-value of <0.05 was considered statistically significant. The findings revealed that NF1 children were more likely to use electronic devices before sleep (72% vs. 34%, *p* = 0.001) and be rocked to sleep (16% vs. 0%, *p* = 0.014), indicating a higher prevalence of potentially disruptive sleep onset behaviors. The time taken to fall asleep did not differ significantly between groups (*p* > 0.05). Children with NF1 were also more likely to nap more frequently (26% vs. 16%, *p* = 0.241) and for longer durations, although differences in nap habits were not statistically significant. Regarding sleep disturbances during the night, children with NF1 exhibited significantly higher rates of nocturnal hyperhidrosis (48% vs. 10%, *p* = 0.002) and restless leg syndrome (22% vs. 4%, *p* = 0.023). The prevalence of bruxism was higher in the NF1 group (48% vs. 28%), but this was not statistically significant (*p* = 0.133). Other disorders such as recurrent nightmares and confusional awakenings were more frequent in NF1 children but did not reach statistical significance (*p* > 0.05). The results regarding daytime manifestations revealed that children with NF1 had higher rates of daytime sleepiness (18% vs. 6%, *p* = 0.076) and cognitive impairments (10% vs. 2%, *p* = 0.162). Behavioral issues, such as mood swings (36% vs. 24%) and aggression/irritability (48% vs. 24%), were more common in the NF1 group, though the *p*-values indicated trends rather than strong significance (*p* > 0.05) (Table 3).

## 4. Discussion

After collecting personal data, we analyzed the presence of pathologies in both the control group, who met the inclusion criterion of having no neurological or neuropsychiatric disorders, and NF1 patients, accounting for any clinical disorders related to the disease. Previous research has shown that sleep disorders can be associated with various underlying conditions or medications used to treat these conditions [1,3]. For example, H1-antihistamines, which act as inverse agonists at the histamine H1 receptor, have been shown to cause drowsiness, fatigue, and cognitive–behavioral deficits due to their impact on the REM sleep latency and duration [18,19]. Similarly, medications for gastroesophageal reflux have not been extensively studied in relation to sleep disorders, but reflux episodes are known to disrupt sleep quality due to sudden awakenings [20,21].

Neuropsychiatric conditions have been strongly correlated with sleep disturbances. Convulsive episodes and seizures are known to fragment sleep, and antiepileptic or antipsychotic medications often exacerbate these issues [22,23]. The inclusion of behavioral habits, such as bedtime routines and electronic device usage, in this study highlights additional contributors to sleep disorders. Excessive screen time has been associated with reduced melatonin secretion, a delayed sleep onset, and poorer sleep quality [24,25]. Our findings confirm that 72% of NF1 children used electronic devices before bedtime, compared to only 34% of controls, aligning with these studies. Additionally, co-sleeping behaviors were frequent in both groups (54% of NF1 children vs. 50% of controls), reflecting parental compliance as a key factor in sleep behaviors [26].

Sleep disorders were significantly more prevalent and frequent in NF1 patients compared to controls, consistent with the prior literature emphasizing a greater burden of sleep issues in children with chronic conditions [27,28]. Disorders at the sleep onset or upon waking were more common in NF1 children, with 20% reporting early nocturnal awakenings compared to 10% of controls. Sleep paralysis affected 12% of NF1 children but was absent in controls, supporting its link to neurological disturbances [29].

Nocturnal awakenings and associated disturbances, such as hyperhidrosis and bruxism, were notably higher in NF1 patients (48% vs. 10% and 48% vs. 28%, respectively), consistent with findings from other studies that identified autonomic dysfunction as a contributor [30,31]. Restless leg syndrome and periodic limb movements were also more prevalent in NF1 children (22% vs. 4%), corroborating findings that suggest a potential link between NF1 and disrupted motor control during sleep [32].

Regarding respiratory sleep disorders, the literature lacks robust data correlating NF1 with conditions like OSA. However, specific features of NF1, such as neurofibromas within the airways, could predispose individuals to OSA [33]. In a recent article [34], among twenty-two patients affected by NF1, ten (45%) patients had no OSA, one (5%) had mild OSA, two had (9%) moderate OSA, and nine (41%) had severe OSA. Notably, none of the patients had central sleep apnea. CPAP was useful, in many of these patients, in avoiding the need for a tracheostomy. This aligns with our findings, where breathing difficulties and OSAS were reported in 16% and 14% of NF1 children, respectively, compared to none of the controls. These disturbances significantly affect sleep quality and daytime functioning [35].

The impact of sleep disorders on daily functioning was evident. While 90% of controls showed no daytime clinical manifestations, only 68% of NF1 patients were unaffected. NF1 patients experienced higher rates of daytime sleepiness (18% vs. 6%), headaches (10% vs. 4%), and reduced cognitive performance (10% vs. 2%), echoing the findings of prior studies [36,37]. Behavioral changes were also more frequent in NF1 children, with 48% of parents reporting increased aggression and irritability (vs. 24% of controls), consistent with the existing literature [38].

Parental emotional involvement showed mixed findings. NF1 mothers experienced higher rates of anxiety/panic attacks (12% vs. 0%) but lower irritability (8% vs. 38%) than controls, reflecting the psychological toll of caregiving. Fathers reported minimal emotional involvement, with 92% unaffected in the NF1 group compared to 64% of controls, consistent with prior studies suggesting mothers often bear a greater emotional burden in managing chronic conditions [39,40].

Finally, melatonin use was reported in both groups but was more frequent in NF1 patients (20% vs. 10%), with 4% of NF1 children finding no benefit. The medication use for sleep disorders was minimal (2% in both groups), indicating limited pharmacological intervention [41].

Statistically, NF1 children were significantly more likely to use electronic devices before sleep (72% vs. 34%, *p* = 0.001) and experience nocturnal hyperhidrosis (48% vs. 10%, *p* = 0.002). Restless leg syndrome was also more frequent in the NF1 group (22% vs. 4%, *p* = 0.023). While mood swings (36% vs. 24%) and aggression/irritability (48% vs. 24%) were higher in the NF1 group, these trends did not reach statistical significance (*p* > 0.05). Daytime sleepiness (18% vs. 6%) and cognitive impairments (10% vs. 2%) were more prevalent in the NF1 group, though not statistically significant. These findings highlight the increased burden of sleep disorders in NF1, emphasizing the need for targeted interventions. Future research should investigate long-term effects and potential management strategies.

The observed differences in sleep metrics (measured using an Apple Watch Ultra) between NF1 patients and controls align with the prior literature highlighting the prevalence of sleep disturbances in children with NF1 [27]. The shorter total sleep duration and reduced sleep efficiency in NF1 children may reflect underlying neurodevelopmental disruptions associated with the disorder [36]. Increased nocturnal awakenings and higher respiratory rates, coupled with lower SpO_2_; levels, suggest a potential contribution of undiagnosed sleep-related breathing disorders, which have been reported in NF1 populations [42]. Reduced REM and deep sleep durations could negatively impact cognitive performance and emotional regulation, as previously documented [43]. Wearable technologies like the Apple Watch Ultra provide an innovative approach for real-time, non-invasive monitoring, bridging gaps in traditional polysomnography and enhancing our understanding of NF1-related sleep pathology. Future research should focus on integrating wearable data with clinical assessments to develop tailored interventions for improving sleep health in this population.

### Strengths and Limitations

This study integrated subjective reports with objective wearable data (Apple Watch Ultra) to offer a multifaceted view of sleep in NF1, capturing both self-reported experiences and physiological indicators of sleep quality. It also underscored how psychosocial factors (e.g., electronic device use, co-sleeping) influence NF1-related sleep disturbances. A comparison with a matched control group without neurological or neuropsychiatric conditions highlighted the specific sleep burden associated with NF1. Aligning these results with established findings on NF1-related respiratory complications and novel insights from wearable technology confirms the study’s relevance and underscores the need for further research.

However, some limitations must be acknowledged. The relatively small sample size may restrict the generalizability of the findings and underscores the value of larger, multicenter investigations. Additionally, the reliance on univariate analyses (chi-square and *t*-tests) limited our ability to adjust for potential confounding factors. Future studies employing multivariable regression models could better quantify the independent relationship between NF1 and sleep disorders.

While parental and self-reported data offer important perspectives, they remain susceptible to reporting and recall biases, particularly regarding nocturnal awakenings and respiratory symptoms. The absence of confirmatory polysomnography means that complex sleep pathologies, including obstructive sleep apnea or periodic limb movement disorders, could not be definitively diagnosed within this study framework. In the present day, real-time monitoring in home settings may be applied to investigate sleep in these patients, reducing the invasiveness and disruption for the patients and providing valuable longitudinal data. Notably, the accuracy of the Apple Watch Ultra as a sleep-monitoring tool in pediatric populations remains to be fully established. In the present study, the device was not calibrated against the gold standard of polysomnography (PSG). Future investigations should include direct comparisons with PSG or reference existing validation studies where available to ensure the reliability of wearable-based sleep metrics in children, particularly those with NF1.

Moreover, the cross-sectional design did not allow for conclusions about causality or temporal changes in sleep patterns over time. Finally, NF1′s broad clinical variability, including the presence of comorbid conditions like seizures or ADHD, may confound specific impacts on sleep unless carefully stratified in future studies.

## 5. Conclusions

The reasons for this higher prevalence are to be found, as previously stated, in the clinical manifestations of the disease and the drugs used to treat the complications. However, studies have recently been conducted on neurofibromin, a protein encoded by the NF1 gene responsible, if mutated, for neurofibromatosis type 1, which demonstrate its direct involvement in the regulation of the circadian rhythm. The studies were conducted by a group of researchers on Drosophila Melanogaster using infrared sensors through which they detected the sleep–wake cycles of insects inside test tubes [41]. What was discovered was that midges generally stay awake for all daytime hours and sleep for about 10 h at night; however, some insects had a totally irregular sleep–wake rhythm [44]. It was precisely the analysis of these ’abnormal’ insects that revealed the involvement of neurofibromin in sleep circuits; they had lesions in an area of the brain known as the ’mushroom body’ in which the expression of NF1 genes was impaired compared to the expression of the same gene in a perfectly normal midge, whose expression increased while awake and decreased while asleep [43]. This study laid the foundation for molecular research to explain a correlation between sleep disorders and NF1.

This study also confirmed the great impact that sleep disorders can have on the lives of sufferers and those around them. This is the main reason why these often-under-diagnosed disorders should be researched more carefully. Proper sleep hygiene is crucial, in fact, for all learning processes and memory. Patients with neurofibromatosis type 1 are already patients who potentially have cognitive–behavioral deficits due to the disease, so correcting sleep disorders could certainly improve these patients’ lives in the personal and social spheres.

For this reason, studies on the association between sleep disorders and NF1 have reason to continue in order to clarify the underlying causes and to plan possible targeted treatments. In fact, most patients with sleep disorders do not undergo medical or behavioral therapy, partly because in most cases one is not even aware that one has a disorder or, if one is, one wrongly tends to underestimate it.

In addition to the factors already assessed, psychological distress and pain—both common in NF1—may independently trigger or worsen sleep disturbances. Without systematically evaluating these confounders, attributing poor sleep solely to NF1 pathology remains challenging. Incorporating validated pain severity scales and psychological inventories in future research would better distinguish disease-specific effects from those arising from comorbid conditions, thus refining both our understanding and clinical management of NF1-related sleep disorders.

In conclusion, our findings reinforce the significant burden of sleep disorders in children with NF1, highlighting the need for targeted interventions. Future research should explore the underlying mechanisms driving these disturbances and develop tailored strategies for improving sleep quality in this vulnerable population.

## Figures and Tables

**Figure 1 biomedicines-13-00907-f001:**
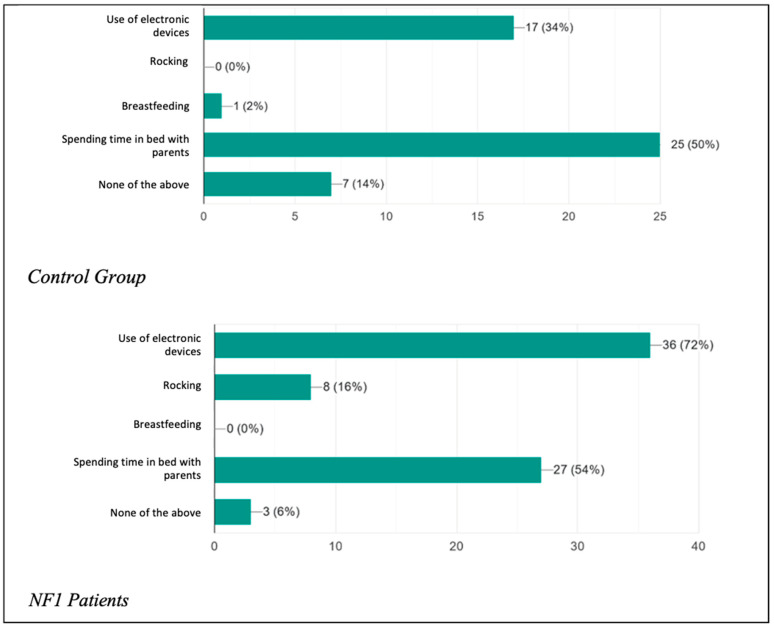
Dependence on certain actions, circumstances, or objects to initiate sleep.

**Figure 2 biomedicines-13-00907-f002:**
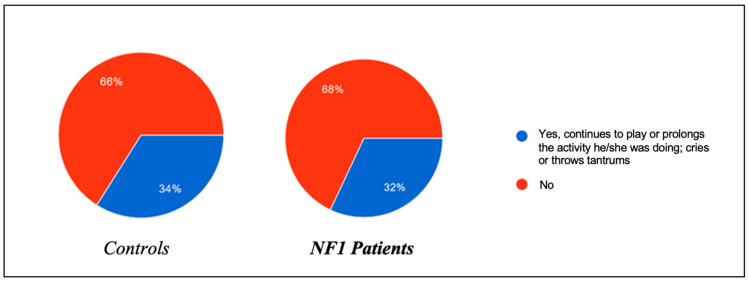
Frequency of behaviors aimed at delaying bedtime in children.

**Figure 3 biomedicines-13-00907-f003:**
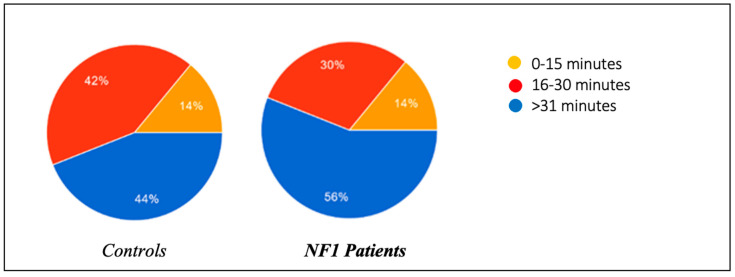
Time taken for children to fall asleep after going to bed.

**Figure 4 biomedicines-13-00907-f004:**
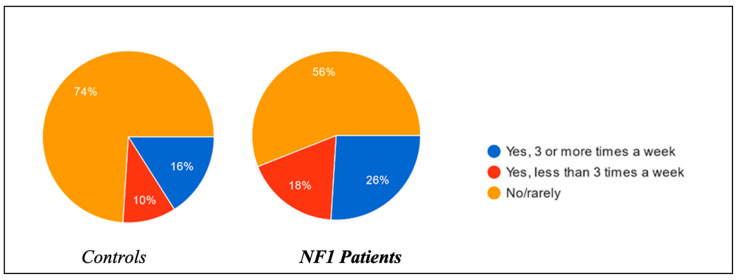
Afternoon naps per week.

**Figure 5 biomedicines-13-00907-f005:**
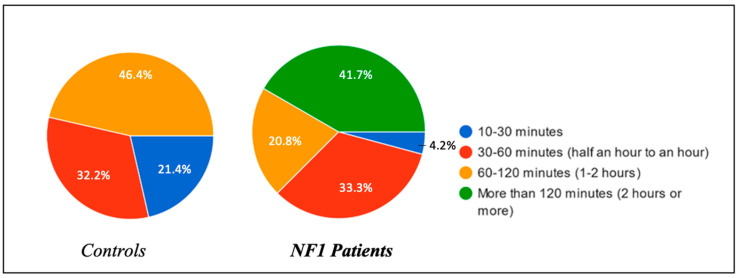
Length of afternoon naps.

**Figure 6 biomedicines-13-00907-f006:**
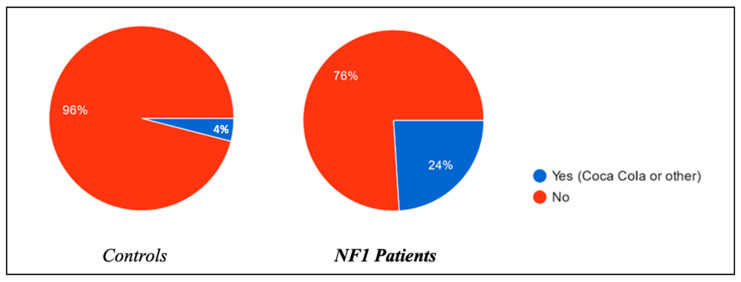
Usage of caffeine among the patients in the afternoon/evening.

**Figure 7 biomedicines-13-00907-f007:**
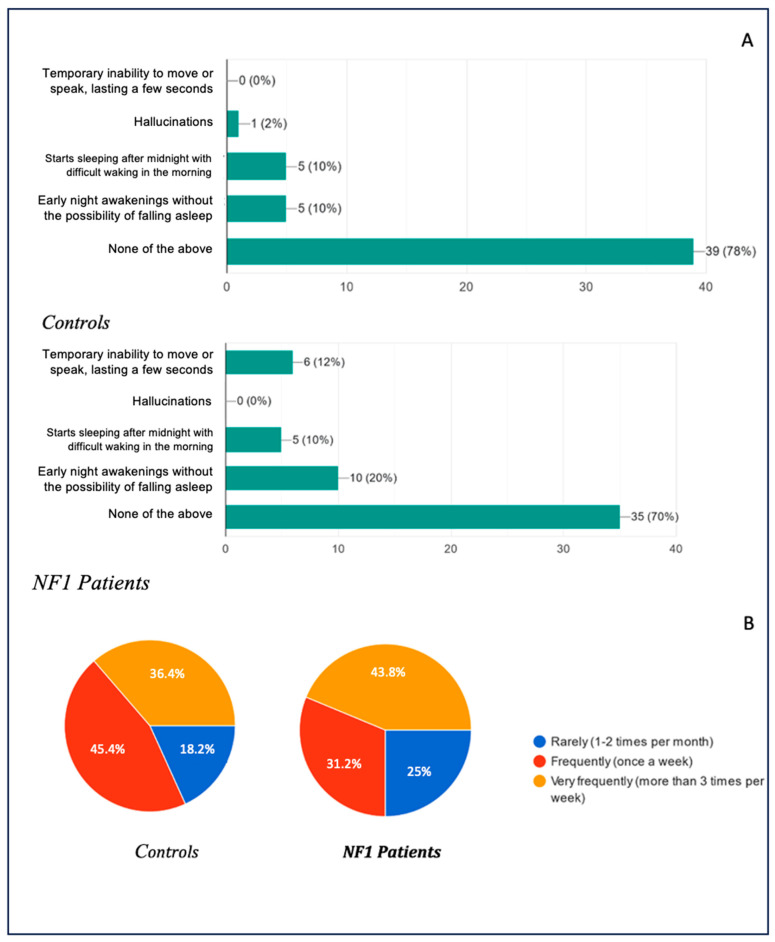
(**A**): Common complaints during sleep onset and waking in children; (**B**): frequency of complaints while falling asleep.

**Figure 8 biomedicines-13-00907-f008:**
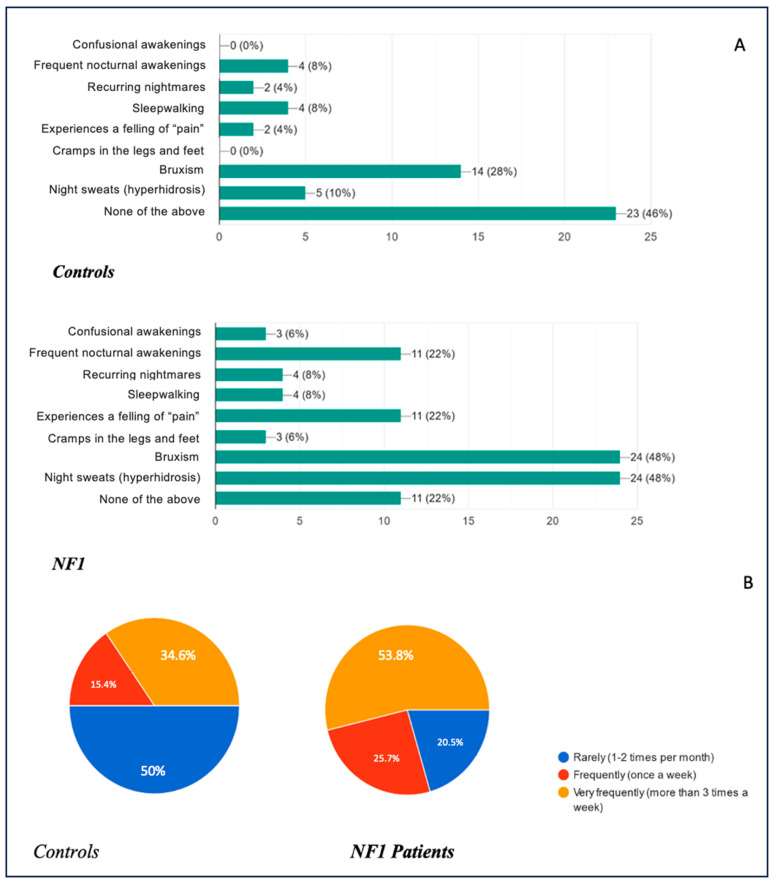
(**A**): Common complaints experienced by children during sleep. (**B**): Frequency of sleep complaints in children.

**Figure 9 biomedicines-13-00907-f009:**
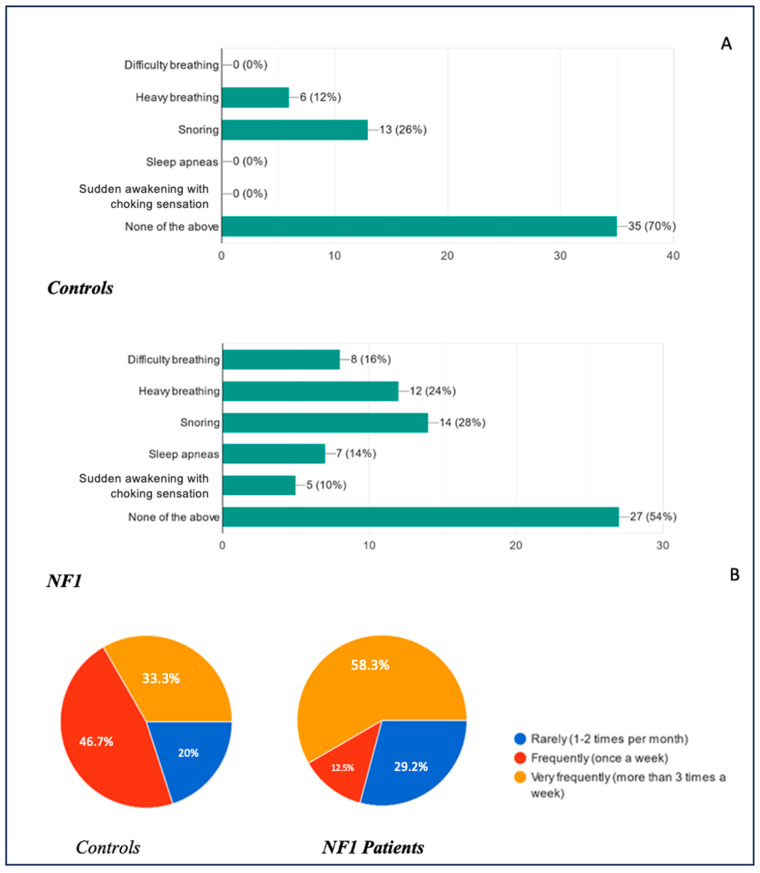
(**A**): Prevalence of respiratory sleep disorders in children; (**B**): frequency of respiratory sleep disorders in children.

**Figure 10 biomedicines-13-00907-f010:**
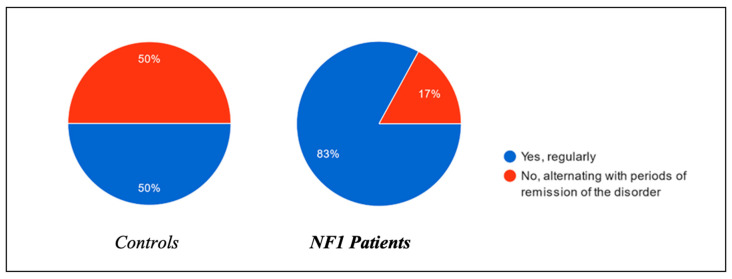
Persistence of sleep disorders in children over time.

**Figure 11 biomedicines-13-00907-f011:**
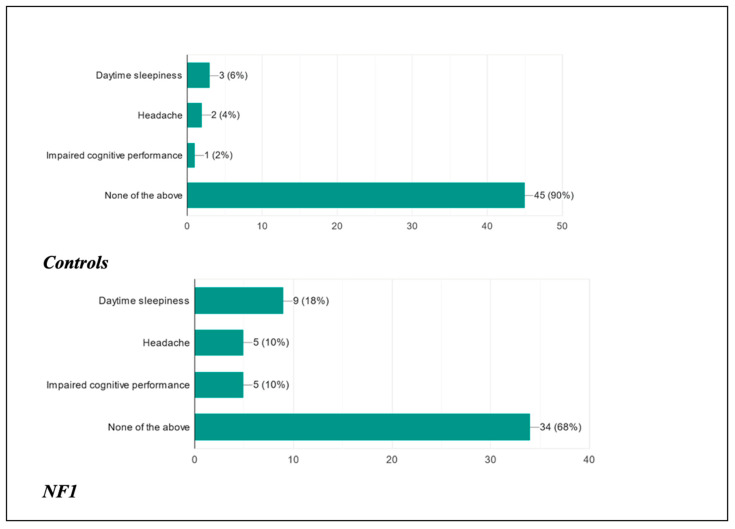
Daytime conditions associated with sleep disorders in children.

**Figure 12 biomedicines-13-00907-f012:**
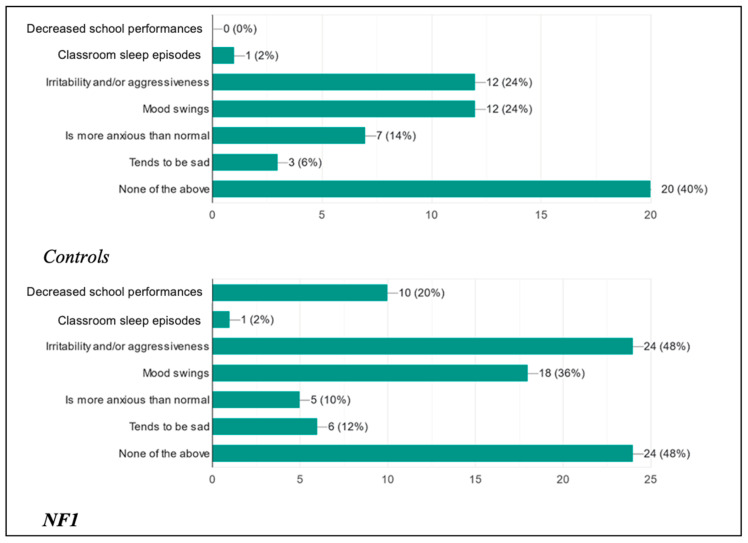
Behavioral changes observed in children after poor sleep.

**Table 1 biomedicines-13-00907-t001:** (**a**) General characteristics of patients in the control group, (**b**) Clinical characteristics of the NF1 group.

(a)							
Patient #	Sex	Age (Years)	Weight (Kg)	Height (m)	BMI	Comorbidities	Drugs
1	F	5	17	1.08	14.57		
2	M	5	20.5	1.07	17.91	ADHD	
3	F	2	14	0.98	14.58		
4	F	7	27	1.3	15.98	Anemia	
5	M	6	18	1.21	12.29		
6	F	9	28	1.34	15.59		
7	M	6	28	1.2	19.44		
8	M	11	39	1.4	19.90	Asthma; ADHD	Inhaled steroids and beta-2 agonists
9	F	10	56	1.45	26.63	Allergic rhinitis	Inhaled steroids
10	F	5	20	1.2	13.89		No
11	M	10	40	1.48	18.26		No
12	F	5	20	1.21	13.66		No
13	M	9	40	1.55	16.65	Allergic rhinitis	Antihistamines
14	M	6	34	1.3	20.12	Asthma	Inhaled beta-2 agonists
15	M	8	28	1.2	19.44	ADHD	No
16	F	11	64	1.5	28.44		No
17	F	11	40	1.4	20.41		No
18	F	12	67	1.5	29.78		No
19	M	8	26	1.32	14.92		No
20	M	11	45	1.55	18.73		No
21	F	6	31	1.37	16.52		No
22	M	2	12	0.86	16.22	Atopic dermatitis	No
23	M	2	9	0.9	11.11	GERD	No
24	M	7	40	1.42	19.84		No
25	M	4	17	1	17.00		No
26	M	3	17	1.02	16.34		No
27	F	6	21	1.2	14.58	Psoriasi	Steroids (topical)
28	M	5	18	1.1	14.88		No
29	M	4	18	1.07	15.72	Asthma	No
30	M	8	32	1.32	18.37		No
31	M	5	16	1.1	13.22		No
32	F	12	60	1.57	24.34		No
33	F	12	60	1.6	23.44	ADHD	No
34	F	7	20	1.15	15.12		No
35	M	5	15	1.02	14.42		No
36	M	5	17	1.1	14.05		No
37	M	8	21	1.12	16.74	GERD, anemia	No
38	M	10	29	1.52	12.55		No
39	F	5	28	1.3	16.57	Atopic dermatitis	No
40	F	11	38	1.5	16.89		No
41	M	9	31	1.3	18.34	Allergic rhinitis	Cetirizine; nasal mometasone
42	F	3	16	0.95	17.73		No
43	F	5	25	1.25	16.00		No
44	F	5	27	1.2	18.75	Allergic rhinitis	Cetirizine; nasal mometasone
45	F	11	32	1.51	14.03		No
46	F	11	34	1.55	14.15		No
47	F	6	22	1.1	18.18		No
48	F	5	18	1.1	14.88		No
49	F	8	30	1.46	14.07		No
50	F	2	15	0.98	15.62	GERD	No
**(b)**							
**Patient #**	**Sex**	**Age (Years)**	**Weight (Kg)**	**Height (m)**	**BMI**	**Comorbidities**	**Drugs**
1	F	12	25	1.3	14.79		
2	F	10	26	1.3	15.38		
3	M	9	28	1.35	15.36	Autism	
4	M	10	30	1.47	13.88		
5	F	9	42	1.36	22.71		
6	F	10	29	1.36	15.68		
7	M	10	34	1.35	18.66		
8	F	9	28	1.3	16.57		
9	F	8	55	1.4	28.06	ADHD	Risperidone and Depakin
10	F	8	18	1.1	14.88	Dysphagia; hypothyroidism	Anti-acid drugs
11	M	8	36	1.21	24.59	Autism	
12	F	8	18	1.2	12.50	ADHD	
13	F	7	21	1.12	16.74		
14	M	7	30	1.2	20.83		
15	M	7	32	1.2	22.22	ADHD	
16	F	7	21	1.26	13.23	GERD and anemia	Alginates
17	F	7	50	1.5	22.22	ADHD	
18	F	8	22	1.26	13.86		Pregabalin
19	M	7	22	1.05	19.95	Epilepsy; dysphagia	Levetiracetam
20	F	7	21	1.26	13.23	Allergic rhinitis, GERD, celiac disease, anemia	Alginates
21	M	6	18	1.2	12.50		
22	M	6	23	1.2	15.97	GERD	
23	M	6	19	1.2	13.19	Atopic dermatitis	Cetirizine; steroids
24	F	5	23	1.1	19.01	Allergic rhinitis	Inhaled steroids
25	F	6	23	1.2	15.97	Allergic rhinitis	Steroids; cetirizine
26	M	11	38	1.5	16.89		
27	M	2	10	0.85	13.84	Dysphagia	
28	M	6	23	1.2	15.97	GERD, autism	
29	M	4	18	1.15	13.61		
30	M	6	22	1.2	15.28		
31	F	5	27	1.25	17.28		
32	M	4	20	1.1	16.53		
33	F	4	20	1.18	14.36		
34	M	5	23	1.2	15.97	Atopic dermatitis	Antisthamines
35	M	7	19	1.2	13.19	Asthma, GERD, anemia, hypercolesterolemia, ADHD	Alginates; inhaled beta-2 agonists
36	F	3	13	0.93	15.03		
37	F	2	8	0.86	10.82	ADHD	
38	F	2	11	0.72	21.22		
39	M	5	17	1.11	13.80		
40	F	5	20	1.18	14.36	Allergic rhinitis, anemia	
41	M	9	38	1.39	19.67		
42	M	7	31	1.15	23.44	Asthma	
43	M	5	20	1.1	16.53		
44	F	4	15	0.98	15.62		
45	F	5	20	1.2	13.89		
46	M	5	18	1.2	12.50	Atopic dermatitis	Topical steroids
47	F	5	12	1.15	9.07		
48	M	3	18	0.96	19.53		
49	M	3	16	1.02	15.38		
50	M	7	30	1.1	24.79	Asthma, GERD, hypertrigliceridemia, ADHD	Alginates; inhaled beta-2 agonists

**Table 2 biomedicines-13-00907-t002:** Statistical comparison between the two groups regarding the questionnaire responses; * *p* < 0.05; ** *p* < 0.01.

Parameter	NF1 Patients	Controls	*p*-Value
Gender (Male/Female)	52%/48%	48%/52%	0.678
Average Age (Years)	6.42 (±2.4)	6.98 (±2.4)	0.231
Co-occurring Diseases (%)	46%	38%	0.512
Taking Medications (%)	24%	14%	0.208
Electronic Device Use Before Sleep (%)	72%	34%	0.001 **
Rocked to Sleep (%)	16%	0%	0.014 *
Co-sleeping with Parents (%)	54%	50%	0.826
No Sleep Onset Habits (%)	6%	14%	0.203
Oppositional Behavior at Bedtime (%)	32%	34%	0.874
Time to Fall Asleep: 0–15 min (%)	56%	44%	0.322
Time to Fall Asleep: 15–30 min (%)	30%	42%	0.267
Time to Fall Asleep: >30 min (%)	14%	14%	1.000
Naps More Than 3x/Week (%)	26%	16%	0.241
Naps Less Than 3x/Week (%)	18%	10%	0.326
No Naps (%)	56%	74%	0.079
Daytime Sleepiness (%)	18%	6%	0.076
Daytime Headaches (%)	10%	4%	0.308
Reduced Cognitive Performance (%)	10%	2%	0.162
Mood Swings (%)	36%	24%	0.288
Aggression/Irritability (%)	48%	24%	0.073
Nighttime Hyperhidrosis (%)	48%	10%	0.002 **
Bruxism (%)	48%	28%	0.133
Restless Leg Syndrome (%)	22%	4%	0.023 *
Confusional Awakenings (%)	6%	0%	0.248
Recurrent Nightmares (%)	8%	4%	0.512
Using Melatonin (%)	20%	10%	0.225
Taking Sleep Medications (%)	2%	2%	1.000

**Table 3 biomedicines-13-00907-t003:** Comparison of the two groups of patients based on the Apple Watch Ultra analysis and the questionnaire (* *p* < 0.05).

Category	Parameter	Measurement Method	NF1 Group (Mean ± SD)	Control Group (Mean ± SD)	Statistical Significance (*p*-Value)
Sleep Duration	Total Sleep Duration (Hours)	Apple Watch	7.2 ± 0.8	8.0 ± 0.6	*p* = 0.001 *
	Time in Bed vs. Time Asleep (Hours)	Apple Watch	8.5 ± 0.7	9.0 ± 0.5	*p* = 0.034 *
	Napping Frequency (Days/Week)	Questionnaire + Apple Watch	2.1 ± 1.4	1.2 ± 0.8	*p* = 0.241
	Napping Duration (Minutes)	Apple Watch	75.4 ± 30.2	38.7 ± 15.6	*p* = 0.013 *
Sleep Stages	REM Sleep Duration (Minutes)	Apple Watch	80 ± 12	95 ± 10	*p* = 0.022 *
	Deep Sleep Duration (Minutes)	Apple Watch	60 ± 10	80 ± 8	*p* = 0.015 *
	Light Sleep Duration (Minutes)	Apple Watch	240 ± 20	270 ± 15	*p* = 0.027 *
Heart Rate	Resting Heart Rate During Sleep (bpm)	Apple Watch	82 ± 5	76 ± 4	*p* = 0.010 *
	Heart Rate Variability (HRV)	Apple Watch	48 ± 6	60 ± 7	*p* = 0.003 *
Respiratory Metrics	Respiratory Rate (Breaths/Minute)	Apple Watch	22 ± 2	18 ± 1.5	*p* = 0.004 *
	Blood Oxygen Levels (SpO_2_;, %)	Apple Watch	95 ± 2	98 ± 1	*p* = 0.002 *
Sleep Efficiency	Sleep Efficiency (%)	Apple Watch	80 ± 5	88 ± 4	*p* = 0.008 *
Sleep Onset	Time to Sleep Onset (Minutes)	Questionnaire + Apple Watch	18.2 ± 7.4	15.4 ± 6.1	*p* > 0.05
Sleep Disturbances	Frequency of Nocturnal Awakenings (Episodes/Night)	Apple Watch	2.4 ± 1.2	0.8 ± 0.6	*p* = 0.003 *
	Nocturnal Hyperhidrosis	Questionnaire	48%	10%	*p* = 0.002 *
	Restless Leg Syndrome Episodes	Questionnaire	22%	4%	*p* = 0.023 *
	Bruxism Episodes	Questionnaire	48%	28%	*p* = 0.133
	Confusional Awakenings	Questionnaire	6%	0%	*p* = 0.047 *
	Recurrent Nightmares	Questionnaire	8%	4%	*p* > 0.05
	Sleep Breathing Disorders (e.g., Apnea Episodes)	Apple Watch	14.2 ± 5.3	6.1 ± 3.7	*p* = 0.021 *
Behavioral and Daytime Effects	Daytime Sleepiness (%)	Questionnaire	18%	6%	*p* = 0.076
	Aggression/Irritability (%)	Questionnaire	48%	24%	*p* > 0.05
	Reduced Cognitive Performance	Questionnaire	10%	2%	*p* = 0.162
Parental Impact	Emotional Impact on Mothers (%)	Questionnaire	76%	46%	*p* = 0.013
	Emotional Impact on Fathers (%)	Questionnaire	92%	64%	*p* = 0.011

## Data Availability

The data presented in this study are openly available in the public repository of the University of Catania [University of Catania, https://www.medclin.unict.it/docenti/martino.ruggieri, accessed on 21 January 2025] and are available upon request to the Corresponding Author.

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
