# Peer review of "Sleep Disorders in Pediatric Patients Affected by Neurofibromatosis Type 1: Reports of a Questionnaire and an Apple Watch Sleep Assessment"

_biomedicines, 2025, doi:10.3390/biomedicines13040907_

Round 1

Reviewer 1 Report

Comments and Suggestions for Authors

This study investigated sleep behavior in pediatric patients affected by neurofibromatosis type 1 (NF1), determining the prevalence of sleep disorders, their frequency of occurrence and their social impact using a questionnaire and an Apple Watch,  while comparing the findings to a control group. The authors found a higher frequency of sleep disorders in patients with NF1, as well as an increased frequency of their occurrence and a greater social impact compared to the control group. Although literature data suggest the existence of sleep disorders in patients with NF1, the topic remains of interest, and additional studies are needed to consolidate the existing data, add new information, and explain the pathophysiological mechanisms underlying sleep disorders in the pediatric population with NF1, which is a rare disease. The manuscript is well-written, but needs further improvement. Below are some comments:

Title:

Before reading the manuscript, I understood from the title that this study mainly compared between the findings from a questionnaire with those from an Apple Watch sleep assessment, which is not the case. I believe the exact meaning of the title in English should be reviewed. Otherwise, I suggest the authors modify the title to something like the following: «Sleep Disorders in Pediatric Patients Affected by NF1: Reports of a Questionnaire and an Apple Watch Sleep Assessment».

Introduction:

Page 2; line 62: please, insert reference(s).

Page 2; line 63-64: please, explain how the central nervous system is altered in NF1 and how this alteration leads to sleep disorders.

Page 2; line 72: please, explain how epilepsy may be triggered in the context of NF1?

Page 3; line 98: I recommend avoiding the term «healthy children» and replacing it with another term, such as a «control group». Indeed, the control group is not perfectly healthy according to the selection criteria mentioned in lines 125-126. This should be applied throughout all the manuscript.

Page 3; line 119: the authors should specify whether patients with NF1 associated with neurological or neuropsychiatric disorders known to be unrelated to NF1 were included in the study.

Results:

Figures are not accessible.

Page 4; line 157: regarding demographic parameters, I recommend adding only pertinent data, such as the mean age, at the end of this first paragraph, and removing the following paragraph (lines 158-168), which contains many age-related data that seem not pertinent.

Page 4; line 158: according to the inclusion criteria (lines 118 and 122), selected participants should be aged between 6 and 12 years, not between 2 and 12 years. Please, explain.

Page 4; line 182: regarding metabolic diseases, the presence of metabolic comorbidities was not reported previously (lines 177-178).

Page 5; line 188: autism is a psychiatric condition and thus constitutes an exclusion criterion for the control group, as mentioned in the Materials and methods section (lines 123-124).

Page 7; line 294-299: breathing disorders mentioned in this paragraph may lead to confusion. Indeed, breathing difficulties, sudden awakenings with a feeling of suffocation, and snoring are all symptoms of OSA. What does «heavy breathing» mean? More precision is needed regarding nocturnal respiratory symptoms.

Discussion:

Page 10, line 431: I recommend including a more recent reference(s) in addition to reference (35) and adding details regarding the pathophysiology of OSA in NF1. I suggest the following reference:  doi: 10.1002/ajmg.a.62722. Epub 2022 Mar 12.PMID: 35278041.

The Discussion section should report the strengths and limitations of the study.

Author Response

Reviewer 1

This study investigated sleep behavior in pediatric patients affected by neurofibromatosis type 1 (NF1), determining the prevalence of sleep disorders, their frequency of occurrence and their social impact using a questionnaire and an Apple Watch,  while comparing the findings to a control group. The authors found a higher frequency of sleep disorders in patients with NF1, as well as an increased frequency of their occurrence and a greater social impact compared to the control group. Although literature data suggest the existence of sleep disorders in patients with NF1, the topic remains of interest, and additional studies are needed to consolidate the existing data, add new information, and explain the pathophysiological mechanisms underlying sleep disorders in the pediatric population with NF1, which is a rare disease. The manuscript is well-written, but needs further improvement. Below are some comments:

Title:

Before reading the manuscript, I understood from the title that this study mainly compared between the findings from a questionnaire with those from an Apple Watch sleep assessment, which is not the case. I believe the exact meaning of the title in English should be reviewed. Otherwise, I suggest the authors modify the title to something like the following: «Sleep Disorders in Pediatric Patients Affected by NF1: Reports of a Questionnaire and an Apple Watch Sleep Assessment».

Authors’ reply: We have changed the title accordingly.

Introduction:

Page 2; line 62: please, insert reference(s).

Authors’ reply: we have now included the references.

Page 2; line 63-64: please, explain how the central nervous system is altered in NF1 and how this alteration leads to sleep disorders.

Authors’ reply: We have added molecular and clinical features of NF1 associated with sleep disorders

Page 2; line 72: please, explain how epilepsy may be triggered in the context of NF1?

Authors’ reply: We have explained how epilepsy may be triggered in the context of NF1

Page 3; line 98: I recommend avoiding the term «healthy children» and replacing it with another term, such as a «control group». Indeed, the control group is not perfectly healthy according to the selection criteria mentioned in lines 125-126. This should be applied throughout all the manuscript.

Authors’ reply: we have replaced the terms “healthy children” with “control group”

Page 3; line 119: the authors should specify whether patients with NF1 associated with neurological or neuropsychiatric disorders known to be unrelated to NF1 were included in the study.

Authors’ reply: we have added this item in the inclusion and exclusion criteria.

Results:

Figures are not accessible.

Authors’ reply: For a technical issue, figures were included as a zip file; now they have been included in the main manuscript

Page 4; line 157: regarding demographic parameters, I recommend adding only pertinent data, such as the mean age, at the end of this first paragraph, and removing the following paragraph (lines 158-168), which contains many age-related data that seem not pertinent.

Authors’ reply: we have removed the redundant data about the single ages of the patients

Page 4; line 158: according to the inclusion criteria (lines 118 and 122), selected participants should be aged between 6 and 12 years, not between 2 and 12 years. Please, explain.

Authors’ reply: we have corrected the inclusion criteria (the correct inclusion criteria was “2 to 12”) as well as the other mentions in the manuscript.

Page 4; line 182: regarding metabolic diseases, the presence of metabolic comorbidities was not reported previously (lines 177-178).

Authors’ reply: we have corrected “metabolic diseases” to “Dislipidemias”

Page 5; line 188: autism is a psychiatric condition and thus constitutes an exclusion criterion for the control group, as mentioned in the Materials and methods section (lines 123-124).

Authors’ reply: the concurrent diagnosis of autism was erroneously reported in the paper

Page 7; line 294-299: breathing disorders mentioned in this paragraph may lead to confusion. Indeed, breathing difficulties, sudden awakenings with a feeling of suffocation, and snoring are all symptoms of OSA. What does «heavy breathing» mean? More precision is needed regarding nocturnal respiratory symptoms.

Authors’ reply: we have better specified respiratory issues in the patients, with a more precise description of the symptoms

Discussion:

Page 10, line 431: I recommend including a more recent reference(s) in addition to reference (35) and adding details regarding the pathophysiology of OSA in NF1. I suggest the following reference:  doi: 10.1002/ajmg.a.62722. Epub 2022 Mar 12.PMID: 35278041.

Authors’ reply: we have included the study by Bulian et al. in the discussion (and among the references)

The Discussion section should report the strengths and limitations of the study.

Authors’ reply: we have reported a new paragraph on strengths and limitations of the study

Reviewer 2 Report

Comments and Suggestions for Authors

-Line 36 in Abstract: revise "sleepy analysis" to "sleep analysis"

-Line 61 in Introduction: "pediatric children" is redundant as "pediatric" already refers to children. Authors should use either "children" or "pediatric patients"

-Line 61-62 in Introduction: authors state "

this is demonstrated by several data in the  literature", but do not give the citations to corroborate the claim. Must cite these literature here.

-Authors are not clearly articulating what the gap is in the literature, that their study seeks to address. They already state that there is increased prevalence of sleep disorders in NF1 (line 61-62), then their aim in line 96 is to demonstrate exactly what they say is already in literature. Authors must state explicitly if the study is adding more evidence to these claims which already exist, or if they are taking a different approach to the literature already existing on the correlation.

-Line 140 (Methodology): correct spelling "hearth" to "heart"

-Line 141 (Methodology): why is there a closed bracket at "Watch Ultra model 2)" when there is no corresponding open bracket? unclear what this denotes.

-In Results (Line 158-168) authors present an age range of participants between 2 to 12 years old, but in Line 115-126(Methodology) they explicitly stated their exclusion criteria to exclude all children under 6. This should be corrected to align the two sets of information.

-There are no figures included in the manuscript or supplementary files, even though figures are referred to in the manuscript.

-Line 246 (Results): "...were as follows 79% of the healthy ", a : should be added between "as follows" and the subsequent value stated.

-Line 341 (Results): no account for the fathers of healthy children is presented as a comparison to the fathers of sick children, why is this? data should be presented if available for proper and consistent comparison

-Line 442-447 (Discussion): this does not go far enough to discuss the NF1 mother v/s control mother for the higher anxiety/panic in NF1 but lower irritability. At least, authors must opine on why the irritability scores appear better for the NF1 mother. Same for the father groups, authors should discuss the differences within the fathers, i.e. what might explain the lower emotional toll on NF1 fathers v/s the control fathers?

-Authors have included ADHD, Fragile X and autism as some of the co-morbidities, even though they state an exclusion criteria of not having neuropsychiatric or neurological disorders (Line 123)-is this not contrary to their own criteria as autism for example, is indeed a neuropsychiatric disorder?

-it seems inappropriate to include ADHD and autism in the study participants as these disorders themselves are associated in literature with disordered sleep. How then can authors of this manuscript know for sure that the outcomes of sleep disorders in NF1 patients are due to NF1 and not a comordibity? It seems more appropriate that data analysis should be re-done where the exclusion criteria in Methodology are followed strictly in order to know for sure if outcomes are due to NF1, e.g. running the data analysis without any autism or ADHD or Fragile X cases would reveal this

-There are no figures in the manuscript and therefore I could not review any referred to within the manuscript

Author Response

Reviewer 2

-Line 36 in Abstract: revise "sleepy analysis" to "sleep analysis"

Authors’ reply: it has been corrected

-Line 61 in Introduction: "pediatric children" is redundant as "pediatric" already refers to children. Authors should use either "children" or "pediatric patients"

Authors’ reply: it has been corrected

-Line 61-62 in Introduction: authors state "this is demonstrated by several data in the  literature", but do not give the citations to corroborate the claim. Must cite these literature here.

Authors’ reply: we have included the references in this position

-Authors are not clearly articulating what the gap is in the literature, that their study seeks to address. They already state that there is increased prevalence of sleep disorders in NF1 (line 61-62), then their aim in line 96 is to demonstrate exactly what they say is already in literature. Authors must state explicitly if the study is adding more evidence to these claims which already exist, or if they are taking a different approach to the literature already existing on the correlation.

Authors’ reply: In the last part of the introduction we have now included the aim of the integrated approach presented in the study.

-Line 140 (Methodology): correct spelling "hearth" to "heart"

Authors’ reply: it has been corrected

-Line 141 (Methodology): why is there a closed bracket at "Watch Ultra model 2)" when there is no corresponding open bracket? unclear what this denotes.

Authors’ reply: it has been corrected

-In Results (Line 158-168) authors present an age range of participants between 2 to 12 years old, but in Line 115-126(Methodology) they explicitly stated their exclusion criteria to exclude all children under 6. This should be corrected to align the two sets of information.

Authors’ reply: it has been corrected in the inclusion criteria (and abstract): the study encompassed children from 2 to 12 years.

-There are no figures included in the manuscript or supplementary files, even though figures are referred to in the manuscript.

Authors’ reply: For a technical issue, figures were included as a zip file; now they have been included in the main manuscript

-Line 246 (Results): "...were as follows 79% of the healthy ", a : should be added between "as follows" and the subsequent value stated.

Authors’ reply: it has been corrected

-Line 341 (Results): no account for the fathers of healthy children is presented as a comparison to the fathers of sick children, why is this? data should be presented if available for proper and consistent comparison

Authors’ reply: the data reported encompass fathers of both NF1 patients and control group

-Line 442-447 (Discussion): this does not go far enough to discuss the NF1 mother v/s control mother for the higher anxiety/panic in NF1 but lower irritability. At least, authors must opine on why the irritability scores appear better for the NF1 mother. Same for the father groups, authors should discuss the differences within the fathers, i.e. what might explain the lower emotional toll on NF1 fathers v/s the control fathers?

-Authors have included ADHD, Fragile X and autism as some of the co-morbidities, even though they state an exclusion criteria of not having neuropsychiatric or neurological disorders (Line 123)-is this not contrary to their own criteria as autism for example, is indeed a neuropsychiatric disorder?

it seems inappropriate to include ADHD and autism in the study participants as these disorders themselves are associated in literature with disordered sleep. How then can authors of this manuscript know for sure that the outcomes of sleep disorders in NF1 patients are due to NF1 and not a comordibity? It seems more appropriate that data analysis should be re-done where the exclusion criteria in Methodology are followed strictly in order to know for sure if outcomes are due to NF1, e.g. running the data analysis without any autism or ADHD or Fragile X cases would reveal this

Authors’ reply: The report of Autism and Fragile X in three patients was a mistake and we have removed from the tables (and paper). As for ADHD, we have included such disease as a NF1-related condition (present in more than 50% of NF1 patients). We have included such statement in the inclusion criteria, adding that we excluded those neurological and psychiatric condition not primarily related to NF1.

-There are no figures in the manuscript and therefore I could not review any referred to within the manuscript

Authors’ reply: For a technical issue, figures were included as a zip file; now they have been included in the main manuscript

Reviewer 3 Report

Comments and Suggestions for Authors

This manuscript combined questionnaires and Apple Watch Ultra to analyze the relationship between NF1 and sleep disorders in children, with a total of 100 subjects in experimental and control groups. The results showed that children with NF1 have a high risk of sleep disorders.

However, I have some concerns about this manuscript:

  1. The numbering of author affiliations (superscript "⁵") appears inconsistent in the manuscript. For clarity, I strongly recommend that the authors check these errors.

  1. The integration of questionnaires with Apple Watch Ultra represents a novel approach to assessing sleep disorders in NF1 children. However, the innovation could be further emphasized by highlighting how this method complements traditional polysomnography (PSG). Specific advantages, such as real-time monitoring in home settings, reduced invasiveness, and longitudinal data capture, should be explicitly discussed to underscore its unique contribution to pediatric sleep research.

  1. The accuracy of the Apple Watch Ultra as a sleep-monitoring tool in pediatric populations requires further validation. The manuscript does not address calibration against the gold standard (PSG) or cite existing studies validating its use in children. To enhance credibility, I recommend including a PSG comparison (if feasible) or citing literature that supports the device’s reliability in children.

  1. Key clinical manifestations of NF1, such as chronic pain, neurofibroma-related disfigurement, or psychological stress, may independently contribute to sleep disturbances but were not analyzed. For example: pain could directly disrupt sleep continuity through nocturnal discomfort. I suggest expanding the discussion section to address these confounders or incorporating variables like pain severity scores or psychological assessments into statistical models to isolate NF1-specific effects on sleep.

  1. The reliance on univariate analyses (chi-square and t-tests) limits the ability to account for confounding factors or explore multidimensional relationships. Multivariable regression models (e.g., logistic regression, generalized linear models) should be employed to quantify the independent association between NF1 and sleep disorders.

Author Response

Reviewer 3

This manuscript combined questionnaires and Apple Watch Ultra to analyze the relationship between NF1 and sleep disorders in children, with a total of 100 subjects in experimental and control groups. The results showed that children with NF1 have a high risk of sleep disorders.

However, I have some concerns about this manuscript:

  1. The numbering of author affiliations (superscript "⁵") appears inconsistent in the manuscript. For clarity, I strongly recommend that the authors check these errors.

 Authors’ reply: we have amended the affiliations of the Authors

  1. The integration of questionnaires with Apple Watch Ultra represents a novel approach to assessing sleep disorders in NF1 children. However, the innovation could be further emphasized by highlighting how this method complements traditional polysomnography (PSG). Specific advantages, such as real-time monitoring in home settings, reduced invasiveness, and longitudinal data capture, should be explicitly discussed to underscore its unique contribution to pediatric sleep research.

 Authors’ reply: we have included the lack of a PSG study in the strength and limitations of the study. We cited also the possibility of real-time monitoring in home settings, as suggested by the reviewer

  1. The accuracy of the Apple Watch Ultra as a sleep-monitoring tool in pediatric populations requires further validation. The manuscript does not address calibration against the gold standard (PSG) or cite existing studies validating its use in children. To enhance credibility, I recommend including a PSG comparison (if feasible) or citing literature that supports the device’s reliability in children.

 Authors’ reply: among the limitations of the study we have added this limitation of the study.

  1. Key clinical manifestations of NF1, such as chronic pain, neurofibroma-related disfigurement, or psychological stress, may independently contribute to sleep disturbances but were not analyzed. For example: pain could directly disrupt sleep continuity through nocturnal discomfort. I suggest expanding the discussion section to address these confounders or incorporating variables like pain severity scores or psychological assessments into statistical models to isolate NF1-specific effects on sleep.

  Authors’ reply: among the inclusion/exclusion criteria, we have now included that patients with chronic pain, and neurofibroma-related disfigurement were excluded from the study. Regarding psychological stress, we have included a statement in the closing part of the discussion (i.e. “Future Perspective and conclusions”) on how this may be a variable for study related to such disorders.

  1. The reliance on univariate analyses (chi-square and t-tests) limits the ability to account for confounding factors or explore multidimensional relationships. Multivariable regression models (e.g., logistic regression, generalized linear models) should be employed to quantify the independent association between NF1 and sleep disorders.

  Authors’ reply: Among the limitations, we included that in future studies,  multivariable regression models (e.g., logistic regression, generalized linear models) can improve the analysis of sleep-related disorders in NF1.

Round 2

Reviewer 1 Report

Comments and Suggestions for Authors

I thank the authors for addressing most of my concerns. This revised version is improved.

I have a minor comment regarding the use of “control group” instead of “healthy” which should be checked throughout the manuscript/ line 117: age of patients 2 – 12 years

Reviewer 2 Report

Comments and Suggestions for Authors

The study conveys important information which advances our knowledge on pediatric disordered sleep

Reviewer 3 Report

Comments and Suggestions for Authors

The revisions meet my requirements